# The Importance of Alpha-Actinin Proteins in Platelet Formation and Function, and Their Causative Role in Congenital Macrothrombocytopenia

**DOI:** 10.3390/ijms22179363

**Published:** 2021-08-29

**Authors:** Leanne R. O’Sullivan, Mary R. Cahill, Paul W. Young

**Affiliations:** 1School of Biochemistry & Cell Biology, University College Cork, T12 XF62 Cork, Ireland; l.r.osullivan@umail.ucc.ie; 2Department of Haematology and CancerResearch@UCC, Cork University Hospital, University College Cork, T12 XF62 Cork, Ireland; MaryR.Cahill@hse.ie

**Keywords:** actin cytoskeleton, actinin, actinin-1, actinin-4, ACTN1, platelets, congenital macrothrombocytopenia, macrothrombocytopenia, CMTP, megakaryocytes

## Abstract

The actin cytoskeleton plays a central role in platelet formation and function. Alpha-actinins (actinins) are actin filament crosslinking proteins that are prominently expressed in platelets and have been studied in relation to their role in platelet activation since the 1970s. However, within the past decade, several groups have described mutations in ACTN1/actinin-1 that cause congenital macrothrombocytopenia (CMTP)—accounting for approximately 5% of all cases of this condition. These findings are suggestive of potentially novel functions for actinins in platelet formation from megakaryocytes in the bone marrow and/or platelet maturation in circulation. Here, we review some recent insights into the well-known functions of actinins in platelet activation before considering possible roles for actinins in platelet formation that could explain their association with CMTP. We describe what is known about the consequences of CMTP-linked mutations on actinin-1 function at a molecular and cellular level and speculate how these changes might lead to the alterations in platelet count and morphology observed in CMTP patients. Finally, we outline some unanswered questions in this area and how they might be addressed in future studies.

## 1. Introduction

A normal platelet count and proper platelet function are essential for haemostasis. Disorders of platelet number of unknown cause are increasingly diagnosed in the haematology laboratory during routine blood counts [1], which may go without further investigation if not associated with bleeding [2], but these findings raise questions about genetic influences on platelet count. The actin cytoskeleton plays critical roles in platelet production by megakaryocytes in the bone marrow, their release into circulation and their function in terms of activation and aggregation in response to vascular damage. Cooperation between myosin motors and crosslinking proteins acting on actin filaments contribute significantly to the dramatic changes in cellular morphology that typify these processes [3]. Actinins are major actin crosslinking proteins in platelets that have recently been linked in genetic studies to abnormalities in platelet count and size [4]—joining other cytoskeletal proteins including myosins (MYH9), fimbrin (another prominent actin crosslinker) and tubulin subunits in this regard [5].

Our objective here is to review what is known about the roles of actinins in platelet generation and function, and outline potential cellular and molecular mechanisms that may underlie the platelet abnormalities associated with actinin-1 mutations. We will first describe the actinin gene family and the structure of actinin proteins before giving a brief perspective on their significance in platelet activation. Then, we will outline the key molecular interactions of actinins with integrins and other platelet proteins. Turning to the genetic variations within the actinin-1 gene that cause macrothrombocytopenia (CMTP), we will focus on the cellular and molecular consequences of these mutations and speculate regarding the stages in platelet formation and function that they may affect. Finally, we will outline some of the open questions that need to be addressed by future research in this area.

## 2. Actinin Genes and Proteins

Actinins are widely expressed, F-actin binding and crosslinking proteins that are part of the spectrin superfamily, which also contains spectrin, utrophin and dystrophin [6,7]. The domain structure of mammalian actinins consists of an amino-terminal actin-binding domain (ABD), a central rod domain and a carboxyl-terminal calmodulin-like (CaM) domain (Figure 1). Actinins function as antiparallel dimers, which allows them to crosslink actin filaments. Dimerization is largely mediated by the central rod domain which places the ABDs at either end of the molecule, spaced approximately 30 nm apart. The actinin dimer has significant elasticity and is curved axially and twisted, allowing it to resist mechanical forces and provide high-affinity binding sites for other proteins [8]. The CaM domain from one subunit can bind to a “neck” region between the ABD and rod of the other subunit in an actinin dimer [9,10]. This close proximity of the CaM and ABD domains from opposing subunits likely explains how Ca^++^ binding to the CaM domain can inhibit the crosslinking of actin filaments in some, Ca^++^-sensitive, actinin isoforms [11] (Figure 1).

There are four genes coding for actinins: ACTN1–4. Actinin-2 and actinin-3 are Ca^++^-insensitive proteins, predominantly expressed in muscle, with some expression of actinin-2 in the brain. By contrast, actinin-1 and actinin-4 can exist as Ca^++^-sensitive or -insensitive isoforms. These isoforms arise due to alternative splicing of two variants of an exon which encodes part of the first EF-hand motif of the CaM domain. Ca^++^-sensitive variants of actinin-1 and actinin-4 are expressed broadly in many tissues [13]. However, the expression of Ca^++^-insensitive variants is somewhat more specific, with Ca^++^-insensitive actinin-1 being expressed in muscle, smooth muscle and some other tissues, while Ca^++^-insensitive actinin-4 was found only in the brain and spinal cord [13].

Platelet actinin was shown to be Ca^++^-sensitive [14] and had probably been assumed in most studies to be the Ca^++^-sensitive, non-muscle isoform of actinin-1. Indeed, the expression of this isoform in platelets was confirmed by cDNA sequencing [15]. However, quantitative proteomics indicates the presence of both actinin-4 and actinin-2 in addition to actinin-1 in platelets, with ~92,000, ~45,000 and ~25,000 copies per platelet estimated for actinin-1, -4 and -2, respectively [16]. We have also detected these three isoforms in platelet lysates by Western blotting with isoform-specific antibodies [13]. Thus, while Ca^++^-sensitive actinin-1 is likely the most abundant platelet actinin isoform, the potential roles of actinin-4 and actinin-2 have not received as much attention as they deserve.

## 3. Actinin in Platelet Activation and Adhesion

Platelet size and shape change are essential components of platelet activation and have been much studied in both the research laboratory and clinical setting. In resting platelets, the actin cytoskeleton resembles spokes of a wheel, with F-actin radiating from the membrane to cell centre and a dense spectrin-rich shell encases the actin core [3]. At the cell centre is a microtubule circumferential band. During activation, cells transition from a discoid to a spherical shape [17]. Actin filaments are severed in this process and then polymerised to form filopodia [17]. Granules are centralised in the platelet during this process [17]. The contraction of the microtubule coil occurs by the action of actin and the myosin IIA which connects microtubules to actin [3]. This fundamental shape change, and the mechanisms that underlie it, are of huge importance clinically [18]. As platelets become activated and undergo shape change, their surface area increases and the available surface for interaction of GPIb/IX and GPIb-IX-V with other platelets and the endothelium is maximised [19]. Hence, the actin cytoskeleton and associated cytoskeletal proteins have key roles in platelet activation.

The involvement of actinin in platelet activation was apparent from early studies describing actinin as an abundant ~100 kDa protein component of the platelet cytoskeleton that is involved in the bundling of actin microfilaments within pseudopods of activated platelets [20,21,22]. Platelet actinin was shown to bundle actin filaments in a Ca^++^-sensitive manner [14], though the potential presence of multiple actinin isoforms in platelets with subtle differences in their Ca^++^ sensitivity was apparent [23]. In the absence of complete amino acid sequences for the various actinins, these studies employed the nomenclature a, b, c based on molecular mass [23]. Two isoforms, dimers aa (2 × 97 kD) and cc (2 × 94 kD), were described as being Ca^++^-sensitive actin-binding and crosslinking proteins, although aa was shown to be more strongly inhibited by Ca^++^ than cc [23]. Additionally, the actin crosslinking efficiency was higher in cc than aa [23]. How these isoforms correspond to subsequently identified actinin-1 and actinin-4 variants and or post translationally modified versions of these is not clear. 

Receptor-mediated activation of platelets with ADP, thrombin or fibrinogen was shown to cause increased association of actinin with the platelet cytoskeleton [24,25,26] and actinin, along with filamin (actin binding protein/ABP), and myosin were proposed to play important roles in the dramatic changes in cell shape associated with platelet activation. The association of actinin with platelet secretory vesicle membranes [27] and its potential to directly interact with membranes containing diacylglycerol in activated platelets [28] were recognized early on, and supported the potential roles of actinin and vinculin [29] in anchoring actin filaments to membranes. 

In surface-activated spread platelets, actinin was localised at punctuate structures and co-localised with actin filaments, in the cytoplasm, at the filopodia edges, on the cell membrane and around the granulomere [30]. Actinin localised with myosin at the cell membrane, in the cytoplasm and near the granulomere and vinculin was also concentrated at punctate structures near the plasma membrane, possibly associated with actinin at sites of adhesion [30]. Actinin has also been localised to structures called actin nodules, which form during early adhesion and spreading of platelets [29,30]. Platelet actin nodules consist of one to four dense F-actin foci which are interconnected by actin fibres and enhance platelet aggregate stability under flow [31]. 

## 4. Molecular Interactions of Actinin in Platelets

### 4.1. Integrins

The major platelet integrin αIIbβ3 plays a significant role in platelet aggregation. There are 80–100,000 copies of integrin αIIbβ3 on platelet plasma membrane and also in alpha granules [32]. A key insight with regard to actinin function in platelets came with the identification of interactions between actinin and integrins via β1 and β3 subunits, and the demonstration of actinin binding to platelet αIIbβ3 integrin [33]. This interaction suggested a potential direct linkage between cell surface integrins and actinin bound to actin filament compared to more indirect mechanisms involving vinculin and talin. 

Regulatory mechanisms that might modulate such integrin:actin linkages via actinin subsequently emerged. For example, it was demonstrated that tyrosine phosphorylation of actinin-1 occurs in phorbol 12-myristate 13-acetate activated platelets and that this phosphorylation also occurs in platelets spreading on fibrinogen and is dependent on αIIbβ3. The kinase responsible for this phosphorylation was identified as focal adhesion kinase (FAK) and the phosphorylation of actinin-1 at tyrosine 12 by FAK was shown to inhibit its binding to actin filaments [15]. Actinin can be dephosphorylated by SHP-1 phosphatase, which occurs in resting and thrombin stimulated platelets [34]. However, in fibrinogen adherent platelets, inactivation of SHP-1 leads to an increase in actinin phosphorylation [34]. In thrombin stimulated platelets, the dephosphorylation of actinin may strengthen its association with integrins and the cytoskeleton, aiding aggregates to resist shear forces [34]. In contrast, fibrinogen activation causes sustained actinin phosphorylation and reduced association with actin, promoting a dynamic network that favours cell movement and reorganisation of actin filaments [34]. These studies indicate potential reciprocal regulation of actinin phosphorylation by FAK and SHP-1 in platelets.

In addition, regulation of actinin by phosphoinositides is likely to be relevant in platelets. Phosphatidylinositol (3,4,5)-trisphosphate (PIP_3_), produced by activated phosphatidylinositol 3-kinase (PI3K), reduces actinin affinity for β integrins and its ability to bundle F-actin, promoting cell detachment at focal adhesions in migrating cells [33,34]. PIP_2_ and PIP_3_ also regulate proteolysis of actinin by calpains, with PIP_2_ decreasing and PIP_3_ increasing cleavage [35]. PIP_3_ increases cleavage by increasing the flexibility of the neck region connecting the ABD and SLR1 of the rod domain, while PIP_2_ stabilises this structure to reduce cleavage [36]. PIP_2_ and PIP_3_ are thus involved in regulating actinin turnover in the cell which controls remodelling of the actin cytoskeleton [37].

Subsequent to inside-out signalling, integrin αIIbβ3 transitions from a low-affinity state and activated integrin molecules are exposed on the platelet surface [3]. Talin and kindlin binding to the cytoplasmic tail of the integrin induce the conformational shift to the high-affinity state, enabling binding of fibrinogen and VWF [1,25]. Further work into the role of actinin in this process has shown that actinin associates with β3 in the resting state in platelets but becomes dephosphorylated and dissociates in response to inside-out signalling from protease activated receptors, with long-lasting activation of αIIbβ3 in response to PAR4 stimulation [38]. Actinin and talin were shown to bind competitively to β3 integrin in fibroblasts adhering to fibronectin, with talin involved in initial adhesion and actinin replacing talin to facilitate adhesion maturation [39]. The expression of the integrin binding domain only in actinin depleted cells reduced force generation on fibronectin, highlighting that actinin also elicits some effect independent of actin interaction [39]. The mechanisms behind this have been assessed using molecular dynamics simulations [40]. The binding of actinin to β3 was shown to cause a proline-induced kink in the β3 transmembrane domain, which impairs integrin activation [40]. It also prevents talin binding to β3, possibly by interacting with both proteins simultaneously [40]. These results are consistent with actinin having a role in setting αIIbβ3 to a low-affinity state, with inside-out signalling causing its dephosphorylation and dissociation, facilitating talin binding and activation of αIIbβ3 [38]. 

β1 integrin also interacts directly with actinin via its cytoplasmic domain [33]. The β1 integrin subunit is found in various platelet receptors, including the fibronectin, laminin and collagen receptors [33]. Notably, talin and actinin bind cooperatively to the β1 integrin cytoplasmic [39] and so the role of actinin in relation to β1 subunit-containing platelet integrins may be different from that for αIIbβ3 [40]. Shams and Mofrad (2017) found that actinin restricts the motion of the β1 cytoplasmic tail, supporting talin association [40]. This regulation of integrins could modify mechanical signal transmission in the processes of adhesion and cell spreading [40].

### 4.2. GPIb-IX-V

Platelet adhesion to the vascular endothelium is an important component of haemostasis and is facilitated mainly by platelet adherence to exposed VWF and collagen via the GPIb-IX-V complex [32]. The GPIb-IX-V complex also interacts with αMβ2 (Mac-1) on leucocytes [41]. The importance of the GPIb-IX-V complex is underscored by the bleeding phenotype exhibited by patients with inherited defects in GPIb—the rare Bernard Soulier syndrome [42]. The VWF–GPIb-IX interaction is a stimulus for platelet aggregation and eventual activation of integrin αIIbβ3 [43]. Actinin and the GPIb-IX complex were co-immunoprecipitated in response to exposure to shear stress, and this was dependent on VWF binding to GPIbα and actinin becoming phosphorylated [44]. Shear stress and the subsequent phosphorylation also enhanced PKN binding to actinin, speculated to indicate a role for actinin in the formation of a signalling complex with GPIb-IX in response to shear and VWF [44]. In resting platelets, non-phosphorylated actinin associates with PI3K, protein kinase N (PKN) and PIP_2_ [43,44]. Shear stress induced the dissociation of actinin and PIP_2_ from PI3K and the production of PIP_3_ [43]. This complex reassembles at GPIb, with actinin linking the receptor to the cytoskeleton and a second phase of PIP_3_ production occurs that is speculated to maintain GPIb in an activated state and stabilise the nascent thrombus [43]. 

### 4.3. CLP36

CLP36 associates with actin filaments in activated platelets via actinin-1 [45]. CLP36 binds to actinin-1 via SLR2 and SLR3 of the rod domain in both resting and activated platelets [45]. The complexed proteins are not localised to focal adhesions, speculated to be due to a regulatory role of CLP36 [45]. This was thought to involve the promotion of actinin–actin interactions at stress fibres through the inhibition of actinin–integrin interactions at focal adhesions, mediated by CLP36 binding [45]. Actinin, non-filamentous actin, CLP36 and plasma membrane calcium ATPase (PMCA) are found in a complex in resting and activated platelets [46]. This demonstrates a potential role for actinin in localising proteins to platelet structures such as filopodia in activated platelets [45].

### 4.4. Regulation of Actinin Molecular Interactions

Phosphorylation of actinin-1 and the regulation of actinin-1 by phosphoinositides and calcium have been mentioned above. Similar regulatory mechanisms have been described for actinin-4, though not in the context of platelets, outlined in Table 1. For example, tyrosine phosphorylation of actinin-4 has also been demonstrated, with phosphorylation at Tyr4 and Tyr31 inhibiting actin binding, whereas phosphorylation at Tyr265 appeared to enhance actin binding as well as calpain cleavage of actinin-4 [47]. Phosphorylation of Tyr4 and Tyr31 was shown to maintain the ABD in a closed conformation, inhibiting actin binding [48]. Tyr4 phosphorylation acts as a switch which regulates phosphorylation of Tyr31, and is absent in calpain cleaved actinin [49]. Tyrosine phosphorylation at different residues (Tyr11 and Tyr13) required FAK and had minimal effects on actin binding while inhibiting calpain cleavage of actinin-4 [50]. Binding of phosphoinositides PIP_2_ and PIP_3_ could also inhibit cleavage of actinin-4 by m-calpain [51]. Thus, similarly to actinin-1, phosphorylation and phosphoinositide binding can modulate actin binding and protease susceptibility of actinin-4. The significance of these regulatory mechanisms in platelets remains unclear, however.

As mentioned, calcium binding negatively regulates the interaction of actinin-1 and actinin-4 with actin [52]. Calcium signalling is critical in platelet activation and assessing the impact of calcium concentration changes on the interaction of actinin with other cellular proteins in platelets is a topic that merits further study. In thrombin activated platelets, phosphorylated actinin has been reported to immunoprecipitate with transient receptor potential channels, possibly indicating a role in store operated calcium entry, which the authors note is similar to a reported role for muscle actinin in calcium channel regulation in cardiac myocytes [53]. Nitrous oxide (NO) may also regulate actinin-1 in platelets by a transient tyrosine nitration, which is thought to prevent its phosphorylation [54]. These authors propose that tyrosine nitration of actinin, through peroxynitrite formation, may be a modulatory mechanism that inhibits platelet adhesion.

## 5. Involvement of Actinin-1 in Platelet Formation

Platelet formation from megakaryocytes can be considered as a number of sequential steps. While actinins have not been studied extensively in this context, there are several of these steps in which a role for actinin has been described or can be inferred based on our understanding of actinins in other cellular processes.

### 5.1. Megakaryocyte Maturation

Megakaryocytes mature by undergoing endomitosis to increase ploidy [55]. This process is thought be important to generate large quantities of RNA and granule contents for platelets [56]. This occurs through a failure of cytokinesis and involves F-actin, myosin IIA and RhoA [55]. RhoA knockdown caused increased MK ploidy, aberrant proplatelet release and macrothrombocytopenia [55]. Additionally, downregulation of the myosin II heavy chain MYH10 by RUNX1 regulates the failure of cytokinesis during endomitosis through a defect in the actin–myosin II contractile ring [55]. Actinin is known to play a central role in the formation of the contractile ring during cytokinesis in other contexts, working in concert with myosin II (reviewed in [52]). Overexpression of actinin causes a failure in cytokinesis and thus polyploidization [57]. Actinin-1 was shown to be upregulated during megakaryopoiesis and knockdown of actinin-1 in megakaryocytes prevented polyploidisation and cell enlargement, while also causing filamin A levels to be diminished [58,59]. These observations suggest that actinin expression levels and potentially its actin filament crosslinking activity may influence the process of endomitosis during megakaryocyte maturation. 

### 5.2. Proplatelet Formation and Platelet Release

Maturing megakaryocytes move from the haematopoietic niche to the vascular niche within the bone marrow and, at the vascular niche, extend cytoplasmic protrusions called proplatelets through the vessel endothelium into the bloodstream [55]. Platelets are then released from the proplatelet in the bloodstream. Microtubules are the major cytoskeletal components of proplatelets, lining the proplatelet shaft with microtubule sliding, driven by the motor protein dynein, making a major contribution to the proplatelet extension process [60,61]. The microtubules loop back into the shaft at the end of each process, forming a platelet-sized tip [60]. Microtubules and their motor proteins, kinesins, also facilitate the delivery of granules and organelles to the platelets, at proplatelet tips [55]. 

However, the actomyosin network is also important for proplatelet formation. Platelets are formed only at the ends of proplatelets, and actin promotes branching of the proplatelets, yielding a greater number of tips [62]. Although actin has a role in increasing tip number on the proplatelets, actin disruption accelerates proplatelet extension [55]. Regulators of actin polymerisation also play a role in platelet formation, with RhoA, Pak2 and mDia1 affecting proplatelet formation [55]. Myosin IIA mutations also decrease the number of proplatelets formed [55]. While the role of actinins in proplatelet formation and extension has not been studied explicitly, they are likely to be involved on the basis of the many other dynamic cellular contexts in which myosin II motors and actinin crosslinking proteins work together. Additionally, megakaryocytes expressing actinin-1 with CMTP-causing actinin-1 mutations exhibited a reduced number of proplatelet tips and increased tip size, suggesting a role for actinin-1 in regulating this process [63].

As discussed previously, actinin has a role in setting αIIbβ3 to a low-affinity state. The constitutively activated integrin has previously been linked to abnormal proplatelet-like formation in cultured BHK cells [64]. Dysregulated integrin activation is a feature of a number of inherited macrothrombocytopenias. In subtypes of Glanzmann thrombasthaenia, mutations in αIIb and β3 cause constitutive activation of the αIIbβ3 and cause macrothrombocytopenia [64,65]. Increased RhoA signaling through defective filamin A and αIIbβ3 interaction also underlies FLNA-linked macrothrombocytopenia [66]. Potentially, actinin also has a role in proplatelet formation through its interaction with αIIbβ3.

Podosomes are actin-rich structures that have been implicated in platelet formation, whereby podosome clusters connected by the actomyosin network enable the passage of proplatelets through the endothelial barrier during thrombopoiesis [67]. Although similar to platelet actin nodules, they differ as podosomes are larger and require WASp and Arp 2/3 for their formation [31]. They use matrix metalloproteases to degrade the basement membrane and thereby facilitate the release of platelets into the bloodstream [68]. Podosome structure has been best characterised in dendritic cells and macrophages and is comprised of an actin core, associated with protrusion, and an outer ring made up of proteins including vinculin and talin, associated with adhesion [69,70]. The core consists of actin and actin binding proteins, WASP, Arp 2/3, cortactin (predominantly associated with branched actin filaments, at the core centre) and actinin (associated with linear actin filaments at the core periphery) [71]. The core and ring are connected via an actomyosin network and cores on adjacent podosomes are connected via radial actin filaments [70]. Tension generated by the F-actin core and radiating actin network regulates the protrusive and adhesive activity of the podosome clusters [70]. 

Wiskott Aldrich syndrome is an X-linked disorder associated with immunodeficiency and thrombocytopenia involving a congenital mutation in the WAS protein (WASP) [72]. WASP is a component of podosomes and a regulator of actin polymerisation, and mutations are associated with premature platelet release into the bone marrow and microthrombocytopenia [73]. Recent studies have also examined mice with megakaryocyte-specific deficiencies in other proteins linked to podosome formation and function. Loss of Arp2/3 impaired the formation of transendothelial pores and the production of cytoplasmic processes, while loss of MHCIIA affected podosome size, distribution and number and caused scattered transendothelial pores [67]. Deficiencies in actin regulatory proteins twinfilin and cofilin were associated with macrothrombocytopenia, increased MK numbers in the bone marrow and spleen and reduced podosome formation [74]. These studies highlight the potential importance of podosomes for proper platelet production and release from megakaryocytes. Actinin is a key component of podosomes in dendritic cells [71] and we have detected both actinin-1 and actinin-4 in the podosomes of cultured murine megakaryocytes (LOS and PY, unpublished observations). Podosome-mediated protrusion of proplatelets into blood vessels may therefore represent a process in which actinins play a critical role during platelet production in vivo.

### 5.3. Proplatelet Fission

When the proplatelet tips are released into the blood, they may take the form of premature platelets, including preplatelets (2–10 µm) or barbell shaped platelets (two 2 µm platelets which are connected via a microtubule coil) [75,76]. These mature into discoid 2 µm platelets, driven by shear forces during blood flow [55] or at sites such as the lung microvasculature [56]. A role for myosin II in this process has been described whereby shear forces activate myosin II, which then drives cleavage furrow formation and fission, resulting in the generation of normal small platelets [77]. These authors propose that macrothrombocytopenia in MYH9-related diseases arises because myosin II heavy chain mutations abrogate normal myosin activity, preventing cleavage furrow formation and platelet fission. Actinins are responsive to shear stress in platelets [43,44] and implicated in mechanosensory signal transduction in other contexts [78,79,80]. The disease-associated actinin-4 mutant K255E has increased affinity for F-actin and was shown to be insensitive to mechanoregulation [81]. Whether actinin-1 mechanosensing is involved in proplatelet fission remains to be assessed. Actinins are likely to crosslink actin filaments in cleavage furrows formed during proplatelet fission in circulating blood and may be regulated by shear stress. Disruption of such a role could represent a mechanism underlying the macrothrombocytopenia associated with mutations in actinin-1. 

## 6. Genetic Studies Linking Actinin-1 to Platelet Production

Mutations in all four actinin genes have been described [82]. Mutations in ACTN2 and ACTN3 cause cardiomyopathies and alterations in muscle physiology, respectively, in line with their expression patterns. By contrast, despite their broad and overlapping expression patterns, ACTN1 and ACTN4 mutations are associated with diseases affecting a very specific physiological process. ACTN4 mutations cause the kidney disease focal segmental glomerulosclerosis [83] and actinin-4 seems to play a role in renal podocytes that cannot be compensated for by actinin-1 [84]. Most recently, ACTN1 mutations that cause a mild, non-syndromic, form of CMTP have been identified (Online Mendelian Inheritance in Man (OMIM) database link: https://www.omim.org/entry/102575, accessed on 25 August 2021). These mutations as well as other genetic evidence linking actinin-1 to platelet production are discussed below. 

### 6.1. Discovery of CMTP-Causing Actinin-1 Mutations

Genetic linkage studies in Japanese and French families led to the identification in 2013 of the first mutations in actinin-1 that cause CMTP [63,85]. Subsequently, these and additional novel mutations were identified in other patient cohorts. A review in 2017 catalogued 20 mutations [4] and more recent studies have brought the total number of different CMTP-causing actinin-1 mutations identified to approximately 40 [86,87,88,89,90]. With one exception (Luo 21; see below), these are all missense mutations that change a single amino acid within the actinin-1 protein, and all appear to be inherited in a dominant fashion. Mutations affecting all the functional domains of the protein (ABD, rod and CaM domain) have now been described [4]. Thrombocytopenia caused by actinin-1 mutations is mild, with platelet counts of typically 50–130 × 10^9^/L and the presence of enlarged platelets with a platelet diameter that has been reported to be 30% greater than controls [4,63]. Bleeding risk is generally low and no abnormalities beyond platelet count and size have been noted. More recently, it has been noted that some actinin-1 mutations, while causing macrocytosis, are not always associated with thrombocytopenia [86]. Rather than exhaustively cataloging these mutations and the clinical characteristics of affected patients, we will focus here on possible mechanisms by which the mutations could lead to platelet defects. 

### 6.2. Known Cellular and Molecular Consequences of CMTP-Linked Actinin-1 Mutation

To explore how CMTP-linked actinin-1 mutations affect platelet production, Kunishima et al. (2013) expressed mutant actinin proteins in cultured murine megakaryocytes [4,63]. They reported a decrease in the number, but an increase in the size, of proplatelet tips for cells expressing actinin-1 mutants, with some differences in actin organization also noted. The proportion of megakaryocytes forming proplatelets was not altered, pointing toward a defect in the later stages of platelet production. Several groups have also reported that expression of actinin-1 with CMTP-linked mutations in cultured cells (fibroblasts, COS7, CHO cells) causes disorganization of the actin cytoskeleton or differences in the distribution of the mutant actinin-1 compared to the wild type protein [63,85,86,91,92]. By contrast, we did not see gross disruption of the actin cytoskeleton in HeLa cells expressing mutant actinin-1, though we did observe more stable association of mutant actinin-1 with the cytoskeleton [93,94]. These differences may be due to the cell lines used or the degree of overexpression of the mutant proteins.

To examine whether the mutations affect the actin binding and crosslinking activity of actinin-1, we also performed actin co-sedimentation assays. It was found that actinin-1 with a mutation in the actin binding domain (R46Q) had a higher affinity for actin filaments, bundled actin filaments more efficiently and was more stably associated with actin filaments in cells compared to the wild type protein [93]. Other ABD mutations [93], as well as mutations within the rod and CaM domains [94], had similar effects on actin bundling. Mutations within the CaM domain did not dramatically alter Ca^++^ binding but had variable effects on the stability of this domain, possibly explaining alterations in actin binding for these mutant proteins (Figure 1). Overall, these observations suggest a common molecular mechanism related to increased actin filament association for most, if not all, CMTP-linked actinin-1 mutations regardless of their location within the protein. Gain-of-function mutations of this type would explain a phenotype being observed in heterozygous individuals despite the presence of 50% wild type actinin-1 protein, as well as a significant amount of the highly similar actinin-4, and are consistent with the dominant inheritance pattern observed.

### 6.3. Effects of Actinin-1 Expression Levels on Platelet Production

Aside from missense mutations, a couple of genome-wide association studies (GWASs) have linked the ACTN1 gene to altered platelet parameters, possibly as a result of alterations in actinin-1 expression levels rather than affecting its coding sequence. Schick et al. (2016) performed a GWAS for platelet count in a cohort of Hispanic/Latino individuals in the USA and identified the strongest association with a variant (rs117672662) located within a putative megakaryocyte-specific enhancer region in an intron of the *ACTN1* gene [95]. The variant allele showed increased transcriptional activity, but levels of actinin-1 protein were not examined. In an another GWAS for variation in blood cell parameters, two variants that linked ACTN1 to platelet parameters were identified [96]. In this study, the same rare variant mentioned above (rs117672662) was linked to alteration in mean platelet volume and platelet distribution width, while a different, more common variant was associated with mean platelet volume, albeit with a smaller effect size. Finally, a case of ACTN1-linked CMTP involving a frameshift mutation in ACTN1 has recently been reported [90]. This mutation is expected to cause production of a severely truncated protein and decreased levels of intact actinin-1 were reported. This suggests that decreased levels of actinin-1 protein can give rise to a similar platelet phenotype as seen in patients with missense mutations. 

## 7. Unanswered Questions and Future Directions

Despite the considerable literature on the topic outlined above, many questions remain regarding the functions of actinins in platelets and platelet production. One neglected area has been the role of actinins other than actinin-1. Actinin-4 is also present in platelets at perhaps 50% the level of actinin-1 [16]. While these isoforms are similar in terms of actin binding properties, they differ in their regulation by phosphorylation, have some isoform specific interactions and can have different localization patterns in cells. The roles of actinin-4, as well as the importance of smaller quantities of Ca^++^ insensitive actinin-2 in platelets merit further investigation.

It would also be nice to see some of our knowledge on actinin biology translated into useful therapeutics. O’Brien et al. (2019) reported a step in that direction with the description of an actinin peptide that can potentially modulate platelet activation [97]. In their search for short linear motifs (SLiMs) as potential peptide therapeutics, they identified an actinin-1 peptide that binds vinculin. When fused to a cell penetrating TAT sequence, this peptide was capable of modulating platelet aggregation, presumably by disrupting the interaction of vinculin with actinin. It will be interesting to see if further progress can be made in this direction. 

Assuming that increased association with actin filaments is the core mechanism underlying actinin-1-linked CMTP, then key questions are: exactly how, and at what stage(s), does this defect disrupt platelet production? Whatever process is affected must be exquisitely sensitive to the crosslinking activity of actinin-1, since no effects of these actinin-1 mutations have been observed in other cell or tissue types. Given the relative rarity of the disorder, the development of a transgenic mouse model in which one or more of the human CMTP-causing mutations is incorporated into the mouse *Actn1* gene, hopefully recapitulating the essential features of the human condition, will be of great assistance in this regard. Such a model would allow one to examine in detail the earlier steps in megakaryocyte development and maturation that are not readily accessible in patients. 

## Figures and Tables

**Figure 1 ijms-22-09363-f001:**
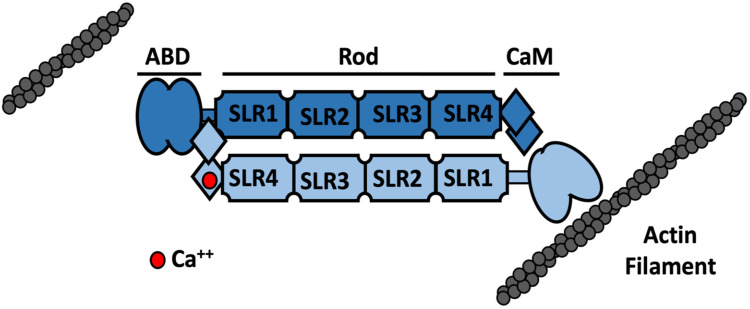
Schematic representation of the structure of the actinin dimer and its regulation by calcium. Actinins are ~100 kDa proteins that form dimers that are ~36 nm long and ~6 nm wide [9]. Each monomer comprises an amino-terminal actin binding domain (ABD) that is made up of two tandem calponin homology (CH1–CH2) domains. These CH domains rearrange from a closed conformation in the absence of actin to a more open, higher affinity state when bound to actin [6,12]. The rod domain facilitates the antiparallel dimerization of actinin and consists of four sprectrin-like repeats (SLRs) with a three-helix bundle structure and high sequence homology to those found in spectrin and dystrophin [6,12]. The carboxyl-terminal calmodulin-like (CaM) domain comprises four helix–loop–helix EF-hand motifs (EF1-4) [7]. EF 3–4 interacts with the neck region between the ABD and first spectrin repeat of the other actinin subunit [9,10]. In Ca^++^-sensitive actinin isoforms, EF1 can bind a single Ca^++^ ion and the solution structure of the actinin-1 CaM domain suggests that while EF3-4 binds to the neck region in the presence of Ca^++^, it is not associated or only loosely associated with the neck region when Ca^++^ is not bound to EF1 [11]. A conformational change involving opening of EF hand 1–2 and structural stabilisation of the domain occurs when bound to Ca^++^ which also reduces the flexibility of the adjacent ABD and inhibits actin binding [11]. The Ca^++^-bound, inhibited state and Ca^++^-free, F-actin-bound states are shown here on the left and right hand sides of a single actinin dimer for illustrative purposes only.

**Table 1 ijms-22-09363-t001:** Regulation of non-muscle actinins by tyrosine phosphorylation.

Isoform	Site	Effect of Phosphorylation	Reference
**Actinin-1**	Tyr 12	Reduced actin binding Enhanced cell mobility Enhanced PKN binding Association with β3 integrin	[15,38,44]
**Actinin-4**	Tyr 4	Inhibits actin binding Regulates Tyr 13 phosphorylation Absent in calpain cleaved actinin-4	[47,48,49]
	Tyr 11	Slight decrease in actin binding Resistance to calpain cleavage	[50]
	Tyr13	Slight decrease in actin binding Resistance to calpain cleavage	[50]
	Tyr 31	Inhibits actin binding	[48,49]
	Tyr 265	Enhances actin binding Increases calpain cleavage of actinin-4	[47]

## Data Availability

Not applicable.

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
