# Peer review of "The Importance of Alpha-Actinin Proteins in Platelet Formation and Function, and Their Causative Role in Congenital Macrothrombocytopenia"

_ijms, 2021, doi:10.3390/ijms22179363_

Round 1

Reviewer 1 Report

This is a very nice and well-written review on an important topic of basic and clinically relevant question : the role of platelet alpha-actinin in health and disease. The authors have an important background in this topic, an "understudied" area of platelet research. This referee has only 3 minor suggestions to be considered by the authors :

a) Perhaps, congenital macrothrombocytopenia (CMTP) could be in the title

b) When they mention/discuss the genetics of actinin mutations, it would be helpful to cite not only the publication(s), but also some genetic databases.

c) Last part of discussion : Could the authors be a little bit more precise about the animal models which would be helpful in the future?

Reviewer 2 Report

This review manuscript by O'Sullivan provides an interesting and comprehensive overview on the role of α-actinins in the production and function of platelets. The manuscript is well written and complete, and could be accepted as is. Some minor additions /corrections might further improve the manuscript.

Figure: "CaM" is spell-check underlined

Phosphorylation of actinins: perhaps it is useful to make a further scheme / table with the location of their individual phosphorylation sites and their effects on the interactions of actinins with other proteins (and consequences on cell function). This might also include some examples of known mutations.

Wiskott-Aldrich syndrome should be briefly introduced. 

The actinins appear to depend on calcium and on phosphorylation in platelets. Is there a mutual influence on the calcium and phosphorylation dependency of actinin? Does phosphorylation influence calcium affinity and does calcium binding influence phosphorylation? The aspect calcium has received less attention than the aspect phosphorylation. Is this justified by the importance of the 2 regulatory mechanisms?
